# HIERARCHICAL ABSTRACTION FOR COMBINATORIAL GENERALIZATION IN OBJECT REARRANGEMENT

**Michael Chang**,[*] **Alyssa L. Dayan, Franziska Meier, Thomas L. Griffiths, Sergey Levine, Amy Zhang**

## ABSTRACT

Object rearrangement is a challenge for embodied agents because solving these tasks requires generalizing across a combinatorially large set of configurations of entities and their locations. Worse, the representations of these entities are unknown and must be inferred from sensory percepts. We present a hierarchical abstraction approach to uncover these underlying entities and achieve combinatorial generalization from unstructured visual inputs. By constructing a factorized transition graph over clusters of entity representations inferred from pixels, we show how to learn a correspondence between intervening on states of entities in the agent's model and acting on objects in the environment. We use this correspondence to develop a method for control that generalizes to different numbers and configurations of objects, which outperforms current offline deep RL methods when evaluated on simulated rearrangement tasks. [1]

## 1 INTRODUCTION

The power of an abstraction depends on its usefulness for solving new problems. Object rearrangement (Batra et al., 2020) offers an intuitive setting for studying the problem of learning reusable abstractions. Solving novel rearrangement problems requires an agent to not only infer object representations without supervision, but also recognize that the same action for moving an object between two locations can be reused for different objects in different contexts.

We study the simplest setting in simulation with pick-and-move action primitives that move one object at a time. Even such a simple setting is challenging because the space of object configurations is combinatorially large, resulting in long-horizon combinatorial task spaces. We formulate rearrangement as an offline goal-conditioned reinforcement learning (RL) problem, where the agent is pretrained on a experience buffer of sensorimotor interactions and is evaluated on producing actions for rearranging objects specified in the input image to satisfy constraints depicted in a goal image.

Offline RL methods (Levine et al., 2020) that do not infer factorized representations of entities struggle to generalize to problems with more objects. But planning with object-centric methods that do infer entities (Veerapaneni et al., 2020) is also not easy because the difficulties of long-horizon planning with learned parametric models (Janner et al., 2019) are exacerbated in combinatorial spaces.

Instead of planning with parametric models, our work takes inspiration from non-parametric planning methods that have shown success in combining neural networks with graph search to generate long-horizon plans. These methods (Yang et al., 2020; Zhang et al., 2018; Lippi et al., 2020; Emmons et al., 2020) explicitly construct a transition graph from the experience buffer and plan by searching through the actions recorded in this transition graph with a learned distance metric. The advantage of such approaches is the ability to stitch different path segments from offline data to solve new problems. The disadvantage is that the non-parametric nature

Figure 1: NCS uses a two-level hierarchy to abstract sensorimotor interactions into a graph of learned state transitions. The affected entity is in black.

---

[*] work done as an intern at Meta AI. Correspondence to: mbchang@berkeley.edu and amyzhang@meta.com

[1] A step-by-step explanatory video of our method can be found in the supplementary material.

of such methods requires transitions that will be used for solving new problems to have already been recorded in the buffer, making conventional methods, which store entire observations monolithically, ill-suited for combinatorial generalization. Fig. 2b shows that the same state transition can manifest for different objects and in different contexts, but monolithic non-parametric methods are not constrained to recognize that all scenarios exhibit the same state transition at an abstract level. This induces an blowup in the number of nodes of the search graph. To overcome this problem, we devise a method that explicitly exploits the similarity among state transitions in different contexts.

Our method, **Neural Constraint Satisfaction** (NCS), marries the strengths of non-parametric planning with those of object-centric representations. Our main contribution is to show that factorizing the traditionally monolithic entity representation into action-invariant features (its **type**) and action-dependent features (its **state**) makes it possible during planning and control to reuse action representations for different objects in different contexts, thereby addressing the core combinatorial challenge in object rearrangement. To implement this factorization, NCS constructs a two-level hierarchy (Fig. 1) to abstract the experience buffer into a graph over state transitions of individual entities, separated from other contextual entities (Fig. 3). To solve new rearrangement problems, NCS infers what state transitions can be taken given the current and goal image observations, re-composes sequences of state transitions from the graph, and translates these transitions into actions.

In §3 we introduce a problem formulation that exposes the combinatorial structure of object rearrangement tasks by explicitly modeling the independence, symmetry, and factorization of latent entities. This reveals two challenges in object rearrangement which we call the **correspondence problem** and **combinatorial problem**. In §4 we present NCS, a method for controlling an agent that plans over and acts with emergent learned entity representations, as a unified method for tackling both challenges. We show in §5 that NCS outperforms both state-of-the-art offline RL methods and object-centric shooting-based planning methods in simulated rearrangement problems.

## 2 Related Work

The problem of discovering re-composable representations is generally motivated by combinatorial task spaces. The traditional approach to enforcing this compositional inductive bias is to compactly represent the task space with MDPs that human-defined abstractions of entities, such as factored MDPs Boutilier et al. (1995; 2000); Guestrin et al. (2003a), relational MDPs Wang et al. (2008); Guestrin et al. (2003b); Gardiol & Kaelbling (2003), and object-oriented MDPs Diuk et al. (2008); Abel et al. (2015). Approaches building off of such symbolic abstractions (Chang et al., 2016; Battaglia et al., 2018; Zadaianchuk et al., 2022; Bapst et al., 2019; Zhang et al., 2018) do not address the problem of how such entity abstractions arise from raw data. Our work is one of the first to learn compact representations of combinatorial task spaces directly from raw sensorimotor data.

Recent object-centric methods (Greff et al., 2017; Van Steenkiste et al., 2018; Greff et al., 2019; 2020; Locatello et al., 2020a; Kipf et al., 2021; Zoran et al., 2021; Singh et al., 2021) do learn entity representations, as well as their transformations (Goyal et al., 2021; 2020), from sensorimotor data, but only do so for modeling images and video, rather than for taking actions. Instead, we study *how well entity-representations can reused for solving tasks*. Kulkarni et al. (2019) considers how object representations improve exploration, but we consider the offline setting which requires zero-shot generalization. Veerapaneni et al. (2020) also considers on control tasks, but their shooting-based planning method in suffers from compounding errors as other learned single-step models do (Janner et al., 2019), while our hierarchical non-parametric approach enables us to plan for longer horizons.

Non-parametric approaches have recently become popular for long horizon planning (Yang et al., 2020; Zhang et al., 2018; Lippi et al., 2020; Emmons et al., 2020; Zhang et al., 2021), but the drawback of these approaches is they represent the entire scenes monolithically, which causes a blowup of nodes in combinatorial task spaces, making it infeasible for these methods to be applied in rearrangement tasks that require generalizing to novel object configurations with different numbers of objects. Similar to our work, Huang et al. (2019) also tackles rearrangement problems by searching over a constructed latent task graph, but they require a demonstration during deployment time, whereas NCS does not because it reuses context-agnostic state transitions that were constructed during training. Zhang et al. (2021) conducts non-parametric planning directly on abstract subgoals rather than object-centric states — while similar, the downside of using subgoals rather than abstract states is that those subgoals are not used to represent equivalent states and therefore cannot generalize to new states at test time. Our method, NCS, captures both reachability between known states and new, unseen states that can be mapped to the same abstract state.

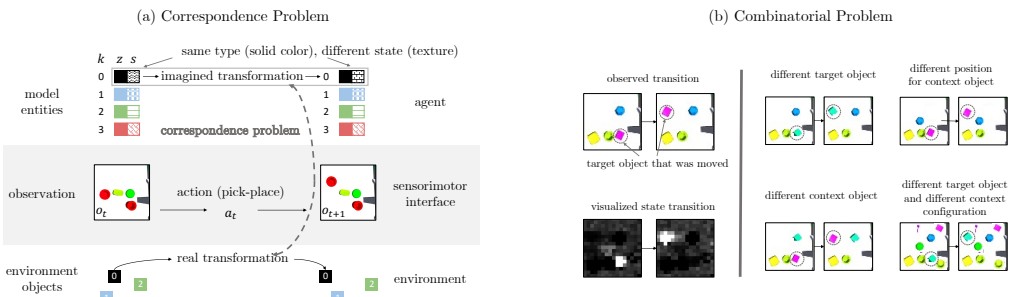

Figure 2: **Solving object rearrangement requires solving two challenges**. (a) The **correspondence problem** is the problem of abstracting raw sensorimotor signal into representations of entities such that there is a correspondence between how an agent intervenes on an entity and how its action affects an object in the environment. $k$ denotes the index of the entity, $z$ denotes its type (shown with solid colors), and $s$ denotes its state (shown with textures). The entity representing the moved object is in black. (b) The **combinatorial problem** is the problem of representing the combinatorial task space in a way that enables an agent to transfer knowledge of a given state transition (indicated by the dotted circle) to different contexts.

## 3 GOAL-CONDITIONED REINFORCEMENT LEARNING WITH ENTITIES

This section introduces a set of modifications to the standard goal-conditioned partially observed Markov decision process (POMDP) problem formulation that explicitly expose the combinatorial structure of object rearrangement tasks of the following kind: "Sequentially move a subset (or all) of the objects depicted in the current observation $o_1$ to satisfy the constraints depicted in the goal image $o_g$." We assume an offline RL setting, where the agent is trained on a buffer of transitions $\{(o_1, a_1, ...a_{T-1}, o_T)\}_{n=1}^N$ and evaluated on tasks specified as $(o_1, o_g)$.

The standard POMDP problem formulation assumes an observation space $\mathcal{O}$, action space $\mathcal{A}$, latent space $\mathcal{H}$, goal space $\mathcal{G}$, observation function $E : \mathcal{H} \to \mathcal{O}$, transition function $P : \mathcal{H} \times \mathcal{A} \to \mathcal{H}$, and reward function $R : \mathcal{H} \times \mathcal{G} \to \mathbb{R}$. Monolithically modeling the latent space this way does not expose commonalities among different scenes, such as scenes that contain objects in the same location or scenes with multiple instances of the same type of object, which prevents us from designing control algorithms that exploit these commonalities to collapse the combinatorial task space.

To overcome this issue, we introduce structural assumptions of independence, symmetry, and factorization to the standard formulation. The *independence* assumption encodes the intuitive property that objects can be acted upon without affecting other objects. This is implemented by decomposing the latent space into independent subspaces as $\mathcal{H} = \mathcal{H}^1 \times ... \times \mathcal{H}^K$, one for each independent degree of freedom (e.g. object) in the scene. The *symmetry* assumption encodes the property that the the same physical laws apply to all objects. This is implemented by constraining the observation function $E$, transition function $P$ and reward function $R$ to be shared across all subspaces, thereby treating $\mathcal{H}^1 = ... = \mathcal{H}^K$. We define an **entity**[2] $h \in \mathcal{H}^k$ as a member of such a subspace, and an **entity-set** as the set of entities $\mathbf{h} = (h^1, ..., h^K)$ that explain an observation, similar to Diuk et al. (2008); Weld. Lastly, the *factorization* assumption encodes that each subspace can be decomposed as $\mathcal{H}^k = \mathcal{Z} \times \mathcal{S}$, where the $z \in \mathcal{Z}$ represents the entity's action-invariant features like appearance, and $s \in \mathcal{S}$ represents its action-dependent features like location. We call $z$ the **type** and $s$ the **state**.

Introducing these assumptions solves the problem of modeling the commonalities among different scenes stated above. It allows us to describe scenes that contain objects in the same location by assigning entities in different scenes to share the same state $s$. It allows us to describe a scene with multiple instances of the same type of object by assigning multiple entities in the scene to share the same type $z$. This formulation also makes it natural to express goals as a set of constraints $\mathbf{h}_g = (h_g^1, ..., h_g^k)$. To solve a task is to take actions that transform the subset of entities in the initial observation $o_1$ whose types are given by $\mathbf{z}_g$ to new states specified by $\mathbf{s}_g$.

Exposing this structure in our problem formulation enables us to exploit it by designing methods that represent entities in an independent, symmetric, and factorized way and that use these three properties to collapse the combinatorial task space. To do so involves solving two problems: the **correspondence problem** of learning to represent entities in this way and the **combinatorial problem**

---

[2]We use "object" to refer to an independent degree of freedom in the environment, and "entity" to refer to the agent's representation of the object.

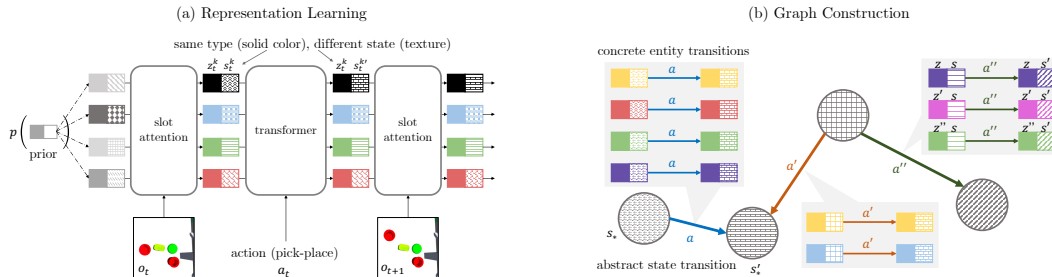

Figure 3: **Modeling**. NCS constructs a two-level abstraction hierarchy to model transitions in the experience buffer. (a) **Level 1:** NCS learns to infer a set of entities from sensorimotor transitions with pick-and-move actions, in which one entity is moved per transition. We enforce that the type $z$ (shown with solid colors) of an entity remains unchanged between time-steps. The GPT dynamics model learns to sparsely predict the states $s$ (shown with textures) of the entities at the next time-step. *This addresses the correspondence problem by forcing the network to use predict and reconstruct observations through the entity bottleneck.* (b) **Level 2:** NCS abstracts transitions over entity-sets into transitions over states of individual entities, constructing a graph where states are nodes and transitions between them are edges. This is done by clustering entity transitions that share similar initial states and final states. *This addresses the combinatorial problem by making it possible for state transitions to reused for different entity types and with different context entities.*

of using these properties to make planning tractable. The correspondence problem is hard because it assumes no human supervision of what the entities are. It also goes beyond problems solved by existing object-centric methods for images and videos because it involves action: it requires representing entities such that there is a correspondence between how the agent models how its actions affect entities and how its actions actually affect objects in the environment. The combinatorial problem goes beyond problems solved by methods for solving object-oriented MDPs, relational MDPs, and factorized MDPs because it requires the agent to recognize whether and how previously observed state transitions can be used for new problems, using learned, not human-defined, entity representations. The natural evaluation criterion for both problems is to test to what extent an agent can zero-shot-generalize to solve rearrangement tasks involving new sets of object configurations that aree disjoint from the configurations observed in training, assuming that the training configurations have collectively covered $\mathcal{Z}$ and $\mathcal{S}$. Our experiments in §5 test exactly this.

**Simplifying assumptions**   To focus on the combinatorial nature of rearrangement, we are not interested in low-level manipulation, so we represent each action as $(w, \Delta w)$, where $w$ are Cartesian coordinates $w = (x, y, z)$. We assume actions sparsely affect one entity at a time and how an action affects an object's state does not depend on its identity. We are not interested in handling occlusion, so we assume that objects are constrained to the $xy$ plane or $xz$ plane and are directly visible to the camera. Following prior work (Hansen-Estruch et al., 2022; Castro et al., 2009), we make a *bisimulation* assumption that the state space can be partitioned into a finite set of equivalence classes, and that there is one action primitive that transitions between each pair of equivalence classes. Lastly, we assume objects can be moved independently. Preliminary experiments suggest that NCS can be augmented to support tasks like block-stacking that involve dependencies among objects, but how to handle these dependencies would warrant a standalone treatment in future work.

## 4   NEURAL CONSTRAINT SATISFACTION

In §3 we introduced a structured problem formulation for object rearrangement and reduced it to solving the correspondence and combinatorial problems. We now present our method, Neural Constraint Satisfaction (NCS) as a method for controlling an agent that plans over and acts with a state transition graph constructed from learned entity representations. This section is divided into two parts: modeling and control. The modeling part is further divided into two parts: representation learning and graph construction. The representation learning part addresses the correspondence problem, while the graph construction and control parts address the combinatorial problem.

### 4.1   MODELING

The modeling component of NCS abstracts the experience buffer into a factorized state transition graph that can be reused across different rearrangement problems. Below we describe how we first

train an object-centric world model to infer entities that are independent, symmetric, and factorized and then construct the state transition graph by clustering entities with similar state transitions. These two steps comprise a two-level abstraction hierarchy over the raw sensorimotor transitions.

**Level 1: representation learning** The first level concerns the unsupervised learning of entity representations that factorizes into their action-invariant features (their **type**) and their action-dependent features (their **state**). Concretely our goal is to model a video transition $o_t, a_t \to o_{t+1}$ as a transition over entity-sets $\mathbf{h}_t, a_t \to \mathbf{h}_{t+1}$, where each entity $h^k$ is factorized as a pair $h^k = (z^k, s^k)$. Given our setting where an action moves only a single object in the environment at a time, successful representation learning implies three criteria: (1) the world model properly identifies the individual entity $h^k$ corresponding to the moved object, (2) only the state $s^k$ of that entity should change, while its type $z^k$ should remain unaffected, and (3) other entity representations $h^{\neq k}$ should also remain unaffected. Criteria (1) and (3) rule out standard approaches that represent an entire scene with a monolithic representation, so we need an object-centric world model instead of a monolithic world model. But criterion (2) rules out standard object-centric world models (e.g. (Veerapaneni et al., 2020; Elsayed et al., 2022; Singh et al., 2022b)), which do not decompose entity representations into action-invariant and action-dependent features.

Because the parameters of a mixture model are independent and symmetric by construction, we propose to construct our factorized object-centric world model as an equivariant sequential Bayesian filter with a mixture model as the latent state, where entity representations are the parameters of the mixture components. Recall that a filter consists of two major components, latent estimation and latent prediction. We implement latent estimation with the state-of-the-art slot attention (SA) (Locatello et al., 2020b), based on the connection Chang et al. (2022) between mixture components and SA slots. We implement latent prediction with the transformer decoder (TFD) architecture (Vaswani et al., 2017) because TFD is equivariant with respect to its inputs. We denote the SA `slots` as $\boldsymbol{\lambda}$ and SA `attn` masks as $\boldsymbol{\alpha}$. We split each `slot` $\lambda \in \mathbb{R}^n$ into two halves $\lambda^z \in \mathbb{R}^{\frac{n}{2}}$ and $\lambda^s \in \mathbb{R}^{\frac{n}{2}}$. Given observations $o$ and actions $a$, we embed the actions as $\tilde{a}$ with an feedforward network and implement the filter as:

$$\hat{\boldsymbol{\lambda}}_1 \sim \text{Gaussian} \qquad \hat{\boldsymbol{\lambda}}^s_{t+1} = TFD\left(\text{queries} = \boldsymbol{\lambda}^s_t, \text{keys/values} = [\boldsymbol{\lambda}^s, \tilde{a}_t]\right)$$

$$\boldsymbol{\lambda}_t, \boldsymbol{\alpha}_t = SA\left(\hat{\boldsymbol{\lambda}}_t, o_t\right) \qquad \hat{\boldsymbol{\lambda}}_{t+1} = \left[\boldsymbol{\lambda}^z_t, \hat{\boldsymbol{\lambda}}^s_{t+1}\right]$$

where $[\cdot, \cdot]$ is the concatenation operator, $\hat{\boldsymbol{\lambda}}$ is the output of the latent prediction step, and $\boldsymbol{\lambda}$ is the output of the latent estimation step. We embed this filter inside the SLATE backbone (Singh et al., 2022a) and call this implementation **dynamic SLATE** (dSLATE).

By constructing $\hat{\boldsymbol{\lambda}}^z_{t+1}$ as a copy of $\boldsymbol{\lambda}^z_t$, dSLATE enforces the information contained $\boldsymbol{\lambda}^z$ to be action-invariant, hence we treat $\boldsymbol{\lambda}^z$ as dSLATE's representation of the entities' types. As for the entities' states, either the action-dependent part of the slots $\boldsymbol{\lambda}^s$ or the attention masks $\alpha$ can be used. Using $\alpha$ may be sufficient and more intuitive to analyze if all objects looks similar and there is no occlusion, while $\boldsymbol{\lambda}^s$ may be more suitable in other cases, and we provide an example of each in the experiments. To simplify notation going forward and connect with the notation in §3, we use $\mathbf{h}$ to refer to $(\boldsymbol{\lambda}, \boldsymbol{\alpha})$, use $z$ to refer to $\boldsymbol{\lambda}^z$, and use $s$ to refer to $\boldsymbol{\lambda}^s$ or $\boldsymbol{\alpha}$. Thus by construction dSLATE satisfies criterion (2). Empirically we observe that it satisfies criterion (1) as well as SLATE does, and that TFD learns to sparsely edit $\boldsymbol{\lambda}^s_t$, thereby satisfying criterion (3).

**Level 2: graph construction** Having produced from the first level a buffer of entity-set transitions $\{\mathbf{h}_t, a_t \to \mathbf{h}_{t+1}\}^N_{n=1}$, the goal of the second level (Fig. 3b) is to use this buffer to construct a factorized state transition graph. The key to solving the combinatorial problem is to construct the edges of this graph to represent not state transitions of entire entity-sets (i.e. $\mathbf{s}_t, a_t \to \mathbf{s}_{t+1}$) as prior work does (Zhang et al., 2018), but state transitions of *individual entities* (i.e. $s^k_t, a_t \to s^k_{t+1}$). Constructing edges over transitions for individual entities rather than entity sets enables the same transition to be reused with different context entities present. Constructing edges over state transitions instead of entity transitions enables the same transition to be reused across entities with different types. This would enable the agent to recompose sequences of previously encountered state transitions for solving new rearrangement problems with different entities in different contexts. Henceforth our use of "state" refers specifically to the state of individual entity unless otherwise stated.

Given our bisimulation assumption that states can be partitioned into a finite number of groups, we construct our graph such that nodes represent equivalence classes among individual states and the edges represent actions that transform a state from one equivalence class to another. To implement

this we cluster state transitions of individual entities in the buffer, which reduces to clustering the states of individual entities before and after the transition. We treat each cluster centroid as a node in the graph, and an edge between nodes is tagged with the single action that transforms one node's state to another's. The algorithm for constructing the graph is shown in Alg. 1 and involves three steps: (1) isolating the state transition of an individual entity from the state transition of the entity-set, (2) creating graph nodes from state clusters, and (3) tagging graph edges with actions.

The first step is to identify which object was moved in each transition, i.e. identifying the entity $h^k$ that dSLATE predicted was affected by $a_t$ in the transition $(\mathbf{h}_t, a_t, \mathbf{h}_{t+1})$. We implement a function isolate that achieves this by solving $k = \arg\max_{k' \in \{1,...,K\}} d(s_t^{k'}, s_{t+1}^{k'})$ to identify the index of the entity whose state has most changed during the transition, where $d(\cdot, \cdot)$ is a distance function, detailed in Table 3 of the Appendix. This converts the buffer of transitions over entity-sets $\mathbf{h}_t, a_t \to \mathbf{h}_{t+1}$ into a buffer of transitions over individual entities $h_t^k, a_t \to h_{t+1}^k$.

The second step is to cluster the states before and after each transition. We implement a function cluster that

---

**Algorithm 1** Building the Graph

1: **input** model, buffer
2: **for** $\{(o_t, a_t, o_{t+1})\}_n$ in buffer **do**
3:     # infer entities from transition
4:     $\{(\mathbf{h}_t, a_t, \mathbf{h}_{t+1})\}_n \leftarrow$ model $(\{o_t, a_t, o_{t+1}\}_n)$.
5:     # identify which entity changed in transition
6:     $\{(h_t^k, a_t, h_{t+1}^k)\}_n \leftarrow$ isolate $(\{(\mathbf{h}_t, a_t, \mathbf{h}_{t+1})\}_n)$
7: **end for**
8: # partition transitions by clustering entities
9: $\{s_*\}_{m=1}^M \leftarrow$ cluster $(\{(s_t^k, a_t, s_{t+1}^k)\}_{n=1}^N)$
10: # transitions between clusters are edges
11: **initialize** graph with nodes $s_*^{[m]}$, for $m \in [1 : M]$
12: **for each** $\{(h_t^k, a_t, h_{t+1}^k)\}_n$ **do**
13:     # infer cluster assignments
14:     $[i], [j] \leftarrow$ bind $(h_t^k)$, bind $(h_{t+1}^k)$
15:     # tag edge with action $a_t$
16:     graph.edges$[i,j] \leftarrow$ create-edge $\left(s_*^{[i]} \xrightarrow{a_t} s_*^{[j]}\right)$
17: **end for**
18: **return** graph

---

uses K-means to returns graph nodes as the centroids $\{s_*\}_{m=1}^M$ of these state clusters.

The third step is to connect the nodes with edges that record actions that actually were taken in the buffer to transform one state to the next. We implement a function bind that, given entity $h^k$, returns the index $[i]$ of the centroid $s_*$ that is the nearest neighbor to the entity's state $s^k$. For each entity transition $(h_t^k, a_t, h_{t+1}^k)$ we bind entity $h_t^k$ and $h_{t+1}^k$ to their associated nodes $s_*^{[i]}$ and $s_*^{[j]}$ and create an edge between $s_*^{[i]}$ and $s_*^{[j]}$ tagged with action $a$, overwriting previous edges based on the assumption that with a proper clustering there should only be one action per pair of nodes.

In our experiments both cluster and bind use the same distance metric (see Table 2 in the Appendix), but other clustering algorithms and distance metrics can also be used. Our experiments (Fig. 11) also show that it is also possible to have more than one action primitive per pair of nodes as long as these actions all map between states bound to the same pair of nodes.

## 4.2 CONTROL

To solve new rearrangement problems, we re-compose sequences of state transitions from the graph. Specifically, the agent decomposes the rearrangement problem into a set of per-entity subproblems (e.g. initial and goal positions for individual objects), searches the transition graph for a transition that transforms the current entity's state to its goal state, and executes the action tagged with this transition in the environment. This problem decomposition is possible because the transitions in our graph are constructed to be agnostic to type and context, enabling different rearrangement problems to share solutions to the same subproblems. The core challenge in deciding which transitions to compose is in determining which transitions are *possible* to compose. That is, the agent must determine which nodes in the graph correspond to the given goal constraints and which nodes correspond to the entities in the current observation, but the current entities $\mathbf{h}_t$ and

---

**Algorithm 2** Action Selection

1: **given** model, graph
2: **input** goal $o_g$, observation $o_t$
3: # infer goal constraints and current entities
4: $\mathbf{h_g}, \mathbf{h_t} \leftarrow$ model $(o_g)$, model $(o_t)$
5: align entity indices of $\mathbf{h_t}$ with those of $\mathbf{h_g}$
6: $\pi \leftarrow$ align $(\mathbf{h_t}, \mathbf{h_g})$
7: permute indices of $\mathbf{h_t}$ according to $\pi$
8: $\mathbf{h_t} \leftarrow (h_t^{\pi[1]}, ..., h_t^{\pi[K]})$
9: identify $k$th goal constraint to satisfy next
10: $k \leftarrow$ select-constraint $(\mathbf{h_t}, \mathbf{h_g})$
11: infer cluster assignments
12: $[i], [j] \leftarrow$ bind $(h_t^k)$, bind $(h_g^k)$
13: action that transforms node $[i]$ to node $[j]$
14: **return** graph.edges$[i,j]$.action

---

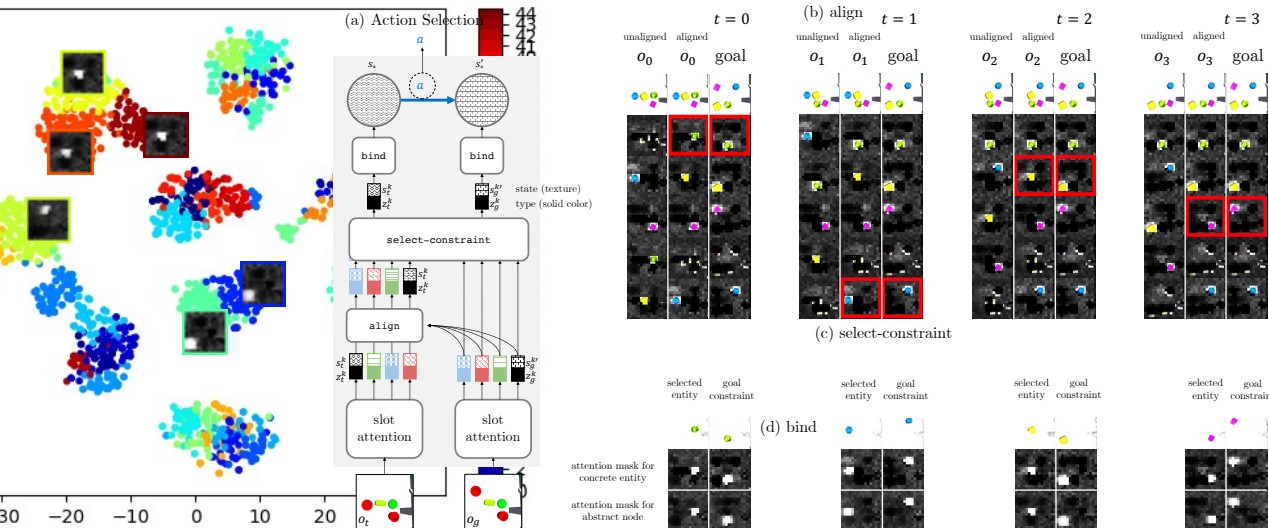

Figure 4: **Planning and control**. Given a rearrangement problem specified only by the current and goal observations $(o_0, o_g)$, NCS decomposes the rearrangement problem into one subproblem $(o_t, o_g)$ per entity. (a) shows the computations NCS uses to solve each subproblem and (b-d) show these steps in context. For each subproblem $(o_t, o_g)$, NCS infers entities from both the current and goal observations. The states of the goal entities indicate constraints on the desired locations of the current entities. (b) NCS aligns the indices of the current entities to those of the goal entities with corresponding types. (c) It selects the index $k$ of the next goal constraint $s_g^k$ to satisfy, as indicated by the red box. The selected goal constraint and current entity are also colored black in (a), and note that their types are the same but states are different; we want to choose the action to transform the state of the current entity to the state of the goal constraint. (d) It binds the selected goal constraint and its corresponding current entity to nodes $s_*$ and $s_*'$ in the transition graph. Lastly, it identifies the edge connecting those two nodes and executes the action tagged to that edge in the environment.

goal constraints $\mathbf{h}_g$ must themselves be inferred from the current and goal observations $o_t$ and $o_g$, requiring the agent to infer both what to do and how to do it purely from its sensorimotor interface.

Our approach takes four steps, summarized in Alg. 2 and Fig. 4. In the first step, we use dSLATE to infer $\mathbf{h}_t$ and $\mathbf{h}_g$ from $o_t$ and $o_g$ (e.g. the positions and types of all objects in the initial and goal images). In the second step (Fig. 4b), because of the permutation symmetry among entities, we find a bipartite matching that matches each entities in $h_g^j$ with a corresponding entity in $h_t^k$ that shares the same type and permute the indices $k$ of $\mathbf{h}_t$ to match those of $\mathbf{h}_g$. We implement a function align that uses the Hungarian algorithm to perform this matching over $(z_t^1, ... z_t^K)$ and $(z_g^1, ... z_g^K)$, with Euclidean distance as the matching cost. The third step selects which goal constraint $h_g^k$ to satisfy next (Fig. 4c). W implement this select-constraint procedure by determining which constraint $h_g^k$ has the highest difference in state with its counterpart $h_t^k$, which reduces to solving the same argmax problem as in isolate with the same distance function used in isolate. The last step chooses an action given the chosen goal constraint $h_g^k$ and its counterpart $h_t^k$, by binding $h_t^k$ and $h_g^k$ to the graph based on their state components and returning the action tagged to the edge between their respective nodes (Fig. 4d). If an edge does not exist between the inferred nodes, then we simply take a random action.

## 5 EXPERIMENTS

We have proposed NCS as a solution to the object rearrangement problem that addresses two challenges: NCS addresses the correspondence problem by learning a factorized object-centric world model with dSLATE and it addresses the combinatorial problem by abstracting entity representations into a queryable state transition graph. Now we test NCS's efficacy in solving both problems.

The key question is whether NCS is better than state-of-the-art offline RL algorithms in generalizing over combinatorially-structured task spaces from perceptual input. As stated in §3, the crucial test for answering this question is to evaluate all methods on solving new rearrangement problems with a disjoint set of object configurations from those encountered during training. The most straightforward way to find a disjoint subset of the combinatorial space is to evaluate with a novel number of

objects. We compare NCS to several offline RL baselines and ablations on two rearrangement environments and find a significant gap in performance between our method and the next best method.

**Environments.** In *block-rearrange* (Fig. 5a), all objects are the same size, shape, and orientation. $\mathcal{S}$ covers 16 locations in a grid. $\mathcal{Z}$ is the continuous space of red-green-blue values from 0 to 1. *robogym-rearrange* (Fig. 5b) is adapte from the OpenAI (2020) rearrange environment and removes the assumptions from *block-rearrange* that all objects have the same size, shape, and orientation. The objects are uniformly sampled from a set of 94 meshes consisting of the YCB object set Calli et al. (2015) and a set of basic geometric shapes, with colors sampled from a set of 13. Although locations are not pre-defined in *robogym-rearrange* as in *block-rearrange*, in practice there is a limit to the number of ways to arrange objects on the table to still be visible to the camera, which makes the bisimulation still a reasonable assumption here. For *block-rearrange* we use the SA attention mask $\alpha$ as the state $s$, and for *robogym-rearrange* we use the action-dependent part of the SA slot $\boldsymbol{\lambda}^s$ as the state $s$.

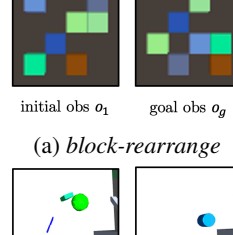

initial obs $o_1$     goal obs $o_g$

(a) *block-rearrange*

initial obs $o_1$     goal obs $o_g$

(b) *robogym-rearrange*

Figure 5: Our environments are *block-rearrange* and *robogym-rearrange*. Fig. 5a shows a complete specification of goal constraints; Fig. 5b shows a partial specification that only specifies the desired locations for two objects.

**Experimental setup.** We evaluate two settings: *complete* and *partial*. In the *complete* setting, the goal image shows all objects in new locations. The *partial* setting is underspecified: only a subset of objects have associated goal constraints (Fig. 5b). In *block-rearrange*, all constraints are unsatisfied in the start state. In *robogym-rearrange*, four constraints are unsatisfied in the start state. Our metric is the *fractional success rate*, the average change in the number of satisfied constraints divided by the number of initially unsatisfied constraints.

The experiences buffer consists of 5000 trajectories showing 4 objects. We evaluate on 4-7 objects for 100 episodes across 10 seeds. Even if we assume full access to the underlying state space, the task spaces are enormous: with $|S|$ object locations and $k$ objects, the number of possible trajectories over object configurations of $t$ timesteps is $\binom{|S|}{k} \times (k \times (|S|-k))^t$, which amounts to searching over more than $10^{16}$ possible trajectories for the complete specification setting of *block-rearrange* with $k = 7$ objects (see Appdx. E for derivation). Our setting of assuming access to only pixels makes the problem even harder.

**Baselines.** The claim of this paper are that, for object rearrangement, (1) object-centric methods fare better than monolithically-structured offline RL methods (2) non-parametric graph search fares better than parametric planning for object rearrangement and (3) a factorized graph search over state transitions of individual entities fares better than a non-factorized graph search over state transitions over entire entity-sets. To test (1), we compare with state-of-the-art pixel-based behavior cloning (BC) and implicit Q-learning (IQL) implementations based off of Kostrikov (2021). To test (2), we compare against a version of object-centric model predictive control (MPC) (Veerapaneni et al., 2020) that uses the cross entropy method over dSLATE rollouts. To test (3), we compare against an ablation (abbrv. NF, for "non-factorized") that constructs a graph with state transitions of entity-sets than of individual states. Our last baseline just takes random actions (Rand).

## 5.1 RESULTS

Figure 1 shows that NCS performs significantly better than all baselines (about a 5-10x improvement), thereby refuting the alternatives to our claims. Most of the baselines perform no better or only slightly better than random. We observe that it is indeed difficult to perform shooting-based planning with an entity-centric world model trained to predict a single step forward (Janner et al., 2019): the MPC baseline performs poorly because its rollouts are poor, and it is significantly more computationally expensive to run (11 hours instead of 20 minutes). We also observe that the NF ablation performs poorly, showing the importance of factorizing the non-parametric graph search. Additional results are in the Appendix.

## 5.2 ANALYSIS

Having quantitatively shown the relative strength of NCS in combinatorial generalization from pixels, we now examine how our key design choices of (1) factorizing entity representations into action-invariant and action-dependent features and (2) querying a state transition graph constructed from action-dependent features contribute to NCS's behavior and performance. Is copying the entity type

Table 1: This table compares NCS with various baselines in the complete and partial evaluation settings of *block-rearrange* and *robogym-rearrange*. The methods were trained on 4 objects and evaluated on generalizing to 4, 5, 6, and 7 objects. We report the fractional success rate, with a standard error computed over 10 seeds.

(a) *block-rearrange*, complete specification.

| Method | 4 | 5 | 6 | 7 |
|---|---|---|---|---|
| **NCS (ours)** | **0.94** ± 0.01 | **0.93** ± 0.00 | **0.93** ± 0.00 | **0.89** ± 0.00 |
| Rand | 0.06 ± 0.02 | 0.07 ± 0.03 | 0.07 ± 0.03 | 0.08 ± 0.03 |
| MPC | 0.16 ± 0.06 | 0.12 ± 0.04 | 0.11 ± 0.04 | 0.10 ± 0.03 |
| NF | 0.07 ± 0.03 | 0.06 ± 0.02 | 0.07 ± 0.02 | 0.08 ± 0.03 |
| IQL | 0.07 ± 0.01 | 0.03 ± 0.00 | 0.02 ± 0.00 | 0.02 ± 0.00 |
| BC | 0.03 ± 0.00 | 0.02 ± 0.00 | 0.01 ± 0.00 | 0.01 ± 0.00 |

(b) *block-rearrange*, complete specification.

| Method | 4 | 5 | 6 | 7 |
|---|---|---|---|---|
| **NCS (ours)** | **0.89** ± 0.01 | **0.86** ± 0.01 | **0.78** ± 0.01 | **0.70** ± 0.01 |
| Rand | 0.06 ± 0.02 | 0.08 ± 0.03 | 0.08 ± 0.03 | 0.08 ± 0.03 |
| MPC | 0.13 ± 0.05 | 0.11 ± 0.04 | 0.10 ± 0.04 | 0.08 ± 0.03 |
| NF | 0.06 ± 0.03 | 0.07 ± 0.03 | 0.08 ± 0.03 | 0.07 ± 0.03 |
| IQL | 0.01 ± 0.01 | 0.07 ± 0.01 | 0.05 ± 0.01 | 0.05 ± 0.00 |
| BC | 0.05 ± 0.01 | 0.04 ± 0.00 | 0.03 ± 0.00 | 0.03 ± 0.00 |

(c) *robogym-rearrange*, complete specification.

| Method | 4 | 5 | 6 | 7 |
|---|---|---|---|---|
| **NCS (ours)** | **0.64** ± 0.01 | **0.47** ± 0.01 | **0.49** ± 0.01 | **0.41** ± 0.01 |
| Rand | 0.01 ± 0.00 | 0.01 ± 0.00 | 0.00 ± 0.00 | 0.00 ± 0.00 |
| MPC | 0.00 ± 0.00 | 0.00 ± 0.00 | 0.00 ± 0.00 | 0.00 ± 0.00 |
| NF | 0.01 ± 0.00 | 0.01 ± 0.00 | 0.00 ± 0.00 | 0.00 ± 0.00 |
| IQL | 0.00 ± 0.00 | 0.00 ± 0.00 | 0.00 ± 0.00 | 0.00 ± 0.00 |
| BC | 0.00 ± 0.00 | 0.00 ± 0.00 | 0.00 ± 0.00 | 0.00 ± 0.00 |

(d) *robogym-rearrange*, partial specification.

| Method | 4 | 5 | 6 | 7 |
|---|---|---|---|---|
| **NCS (ours)** | **0.47** ± 0.01 | **0.33** ± 0.01 | **0.27** ± 0.01 | **0.22** ± 0.01 |
| Rand | 0.005 ± 0.001 | 0.001 ± 0.00 | 0.002 ± 0.001 | 0.001 ± 0.00 |
| MPC | 0.00 ± 0.00 | 0.001 ± 0.001 | 0.00 ± 0.00 | 0.00 ± 0.00 |
| NF | 0.005 ± 0.001 | 0.001 ± 0.00 | 0.002 ± 0.001 | 0.001 ± 0.00 |
| IQL | 0.00 ± 0.00 | 0.00 ± 0.00 | 0.00 ± 0.00 | 0.00 ± 0.00 |
| BC | 0.00 ± 0.00 | 0.00 ± 0.00 | 0.00 ± 0.00 | 0.00 ± 0.00 |

during latent prediction as dSLATE does sufficient for disentangling the location and appearance of objects into the state and type respectively? Does dSLATE learn to sparsely modify only the entity that corresponds to the moved object in the sensorimotor transition, such that the nodes of the state transition graph meaningfully can be reused across entities? These are nontrivial capabilities because NCS is self-supervised on only the experience buffer.

Fig. 4b, which visualizes the `align`, `select-constraint`, and `bind` functions of NCS on *robogym-rearrange*, suggests that, at least for the simplified setting we consider, the answer to both questions is yes. NCS has learned to represent different objects in different slots and construct a graph whose nodes capture location information. Fig. 6 shows a t-SNE (Van der Maaten & Hinton, 2008) plot that clusters entities inferred from the *robogym* environment. Because we have not provided supervision on what states should represent, we observe there are multiple cluster indices that map onto similar groups of points. This reveals that multiple different regions of $\mathcal{S}$ appear to be modeling similar states. We also tried merging redundant clusters, but found that this did not improve quantitative performance.

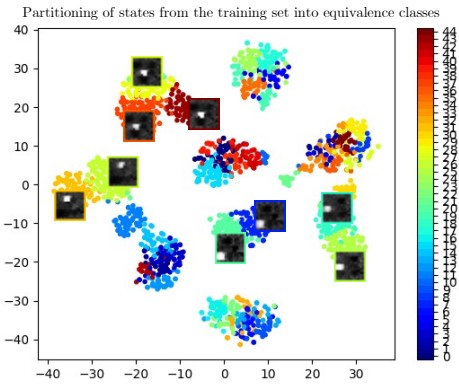

Figure 6: **Nodes as equivalent classes over states.** We show a clustering of states inferred for *robogym-rearrange*, where each cluster centroid is treated as a node in our transition graph. A subset of clusters are labeled with an attention mask computed by averaging the slot attention masks for the entities associated with the cluster.

## 6 DISCUSSION

Object rearrangement offers an intuitive setting for studying how an agent can learn reusable abstractions from its sensorimotor experience. This paper takes a first step toward connecting the world of symbolic planning with human-defined abstractions and the world of representation learning with deep networks by introducing NCS. NCS is a method for controlling an agent that plans over and acts with state transition graph constructed with entity representations learned from raw sensorimotor transitions, without any other supervision. We showed that factorizing the entity representation into action-invariant and action-dependent features are important for solving the correspondence and combinatorial problems that make the object rearrangement difficult, and enable NCS to significantly outperform existing methods on combinatorial generalization in object rearrangement. The implementation of NCS provides a proof-of-concept for how learning reusable abstractions might be done, which we hope inspires future work to engineer methods like NCS for real-world settings.

## ACKNOWLEDGMENTS

This work was done while MC was an intern at Meta AI. We would like to thank Leslie Kaelbling for valuable feedback and Yash Sharma and Yilun Du for valuable discussions. This material is supported in part by the Fannie and John Hertz Foundation, as well as with ONR grant #N00014-18-1-2873.

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
