# OpenReview forum: "Hierarchical Abstraction for Combinatorial Generalization in Object Rearrangement"
_ICLR.cc/2023/Conference — ICLR 2023 poster_

### Official Review · Reviewer_UuMC · 2022-10-24

**Confidence:** 4
**Correctness:** 3
**Technical Novelty And Significance:** 2
**Empirical Novelty And Significance:** Not applicable
**Recommendation:** 6

**Clarity, Quality, Novelty And Reproducibility:**

The paper is currently written in a way that makes it difficult to parse what exactly the main claims and technical contributions are. I felt that Section 2 and Section 3 both introduced new ideas beyond what was summarized in the introduction.

Section 2 purports to provide *a new formulation for goal-conditioned POMDPs*. What exactly about this formulation is *new* with respect to previous literature? References to “standard” goal-conditioned POMDP problem formulation are made but no citations are given. Similarly, mentions of other "standard" POMDP formulations like object-oriented MDPs, relational MDPs, and factorized MDPs are mentioned without citation. My understanding of this section is that it just introduces key “structural assumptions” that are being made about the entities upon which planning is performed?

The paper appears to claim the following as a novel idea: (Section 3) “Our main idea is that generalizing over such a combinatorial space requires the agent to recognize that the same action for moving an object from one location to another can be reused for different objects and in different contexts”. I did not feel as if this claim was properly situated within the context of relevant literature; for example, I would suggest taking a look at [1,2]. Also, in Section 3.1, *"but **our contribution of how we structurally enforce the above three criteria** remains invariant to how the sequential Bayesian filter is implemented, which we expect will advance beyond using SA and GPT"* what exactly is the "contribution" here?

The paper would also benefit from careful editing. There are many sentences that need citations, many typos, and much of the main text is constructed out of long paragraphs with lots of information that are hard to follow. It would also be helpful to clearly define concepts like states, entities, identity, entity-sets, nodes, edges, actions, etc. and provide examples to help explain them early on in the text.



References
=====
1. Aniket Didolkar, Anirudh Goyal, et al. "Neural production systems." Advances in Neural Information Processing Systems 34 (2021): 25673-25687.
2. Goyal, Anirudh, et al. "Object files and schemata: Factorizing declarative and procedural knowledge in dynamical systems." arXiv preprint arXiv:2006.16225 (2020).

### Other suggestions to improve the presentation
- The four tables presented as a Figure (Figure 6) does not look great and was confusing at first.

- *“Not modeling this structure means each scene in the combinatorial space of object configurations is treated as **unrelated** from any other scene”* I think “unrelated” is not correct? A scene-level embedding can still learn commonalities between scenes; however, these representations may be entangled with respect to scene objects?

- Some figures need to be resized to fit the page better, as they look too tiny


**Strength And Weaknesses:**

Strengths
=======
The main high-level idea, which I take to be the proposition of building a higher level of abstraction on top of object-centric representations upon which it is easier to conduct long-term planning --- is solid. NCS appears to be a promising first step based on the decent empirical results on synthetic tasks.

Weaknesses
=======
Given that the key contribution of this paper appears, to me, to be the NCS planner as a whole (specifically, as a demonstration of how object-centric representations can be leveraged for object rearrangement tasks), NCS’s design appears to rely too heavily on certain strong assumptions that cater to the synthetic rearrangement environments it is demonstrated on. I believe this makes it difficult to see how NCS’s technical designs can inform future frameworks applied on more challenging environments. In particular, the treatment of object identity is highly simplistic; NCS copies the identity of each object discovered in the first step forward through time. This requires an assumption of no occlusion and no objects being introduced or leaving during an episode.

I think it's important to discuss and compare against this missing related work [1] --- the question of why non-parametric planning should be conducted over abstract object-centric states versus planning directly on abstract subgoals (which may be less restrictive of an assumption?) is an interesting one.

References
=======
1. Zhang, Lunjun, Ge Yang, and Bradly C. Stadie. "World model as a graph: Learning latent landmarks for planning." International Conference on Machine Learning. PMLR, 2021.

**Summary Of The Paper:**

This paper studies the problem of object rearrangement and focuses particularly on the design of an object-centric planner capable of solving rearrangement tasks requiring multi-step reasoning and systematic generalization. A method, NCS, for combining object-centric representation learning with non-parametric planning is introduced. NCS uses a two-level abstract hierarchy for planning, where a set-structured first level corresponds to sets of entities, and the graph-structured second level corresponds to transitions between “abstracted” entities. Experiments on a few synthetic object rearrangement tasks demonstrate promising performance compared to offline MBRL and object-centric shooting-based planners.


**Summary Of The Review:**

While the main underlying idea is solid and the general research problem is important and relevant to the community, the technical contribution of the paper appears to be of limited significance. Further improvements to the clarity and quality of the paper may reveal otherwise, but this could require significant revisions.

---
Update: After reading the author's response, they have addressed many of the comments and concerns that were brought up. I have increased my score from 5 to 6.

---

> ### Author Response · Authors · 2022-11-15
> **Response**
>
> We thank reviewer UuMC for their helpful comments, which have helped in our paper in many ways, which we have updated and uploaded. We have addressed their writing concerns with several edits to the writing, as indicated by the blue text in the updated manuscript. We address their additional concerns below.
>
> **Assumption of no occlusion**: Copying the identity during the dynamics prediction does not actually require the assumption of no occlusion or the assumption of no objects being introduced or leaving during an episode. This is because the functionality of handling occlusion or the introduction/exit of objects is ultimately the responsibility of the state inference module (which is implemented by slot attention), *not* the dynamics module. The dynamics prediction simply just produces the prior that slot attention updates. Given past evidence that slot-based update rules (e.g. IODINE’s update rule in OP3 https://arxiv.org/pdf/1910.12827.pdf ) can indeed handle occlusion (OP3 Fig 7a) and the re-allocation of slots (Fig 7b), the world model in NCS does not add any additional restrictions that prevent these capabilities.
>
> **Comparison with other non-parametric planning approaches**. Thank you for the reference and we have added a discussion in the related work section of the paper. Our approach does not conflict with methods that plan over subgoals – in fact we state in Section 3.2, that our approach “decomposes the rearrangement problem into a set of per-entity subproblems,” with each entity in the goal image representing a subgoal. Furthermore, our experiments compare against a baseline that treats each set of entities as a monolithic representation, which provides a comparison representative of other monolithic non-parameteric planning approaches.
>
> **Tables in Figure 6**. Thank you for the suggestion and we have fixed their formatting.
>
> **Use of the word unrelated**. Thank you for the suggestion and we have updated the manuscript to clarify what we mean.
>
> **Main claims and contributions**. We have updated the introduction to clarify the main claims and contributions, as well as previewed the main ideas from section 2 and section 3 into the last two paragraphs of the intro.
>
> **New formulation for goal-conditioned POMDPS**. The novelty is the key structural assumptions that we have stated in section 2, which are domain-general. We have added citations to object-oriented MDPs, relational MDPs, and factorized MDPs in Section 2. The difference between our formulation and these prior standard formulations is discussed in the related work section.
>
> **Relation to Neural Production Systems (NPS) and Object Files (SCOFF)**: We thank the reviewer for pointing out these related works and we have added a discussion in the related work section. NPS and SCOFF are great works proposing ways to learn representations of functions that operate on entities. The main difference between our work and NPS/SCOFF is that we tackle different problems: NPS/SCOFF tackle the prediction problem (e.g. video prediction), whereas our work tackles the control problem. The control problem adds a unique and important challenge to learning reusable transformations because it requires the transformations to be useful for accomplishing tasks, whereas this is more difficult to evaluate in a prediction setting. Therefore, the contributions between our work and NPS/SCOFF are largely orthogonal.
>
> **Clarifying “our contribution of how we structurally enforce the above three criteria”**: the contribution is the construction of the two-level hierarchy for modeling as well as the control algorithm described in Figure 4 for planning. This contribution is agnostic to the type of slot-based model that is used.
>
> **Clarify concepts like states, identity, entity-sets, nodes, edges, actions**. We thank the reviewer for pointing this out. These concepts were already explicitly defined in section 2, and we have provided additional clarification with pointers to figures in section 1.

---

> > ### Comment · Reviewer_UuMC · 2022-11-18
> > **Thanks**
> >
> > Thanks for updating the manuscript to address the feedback from reviewers. The new version has improved the readability and presentation a lot.
> >
> > Thanks for adding the recommended references and clarifying similarities/differences to these papers.
> > Also, thanks for clarifying the point about identity copying. I agree that this approach does not add additional restrictions compared to similar slot-based methods such as OP3.
> >
> > I have increased my score from 5 to 6 in light of these changes.

---

### Official Review · Reviewer_wsHP · 2022-10-27

**Confidence:** 4
**Correctness:** 2
**Technical Novelty And Significance:** 2
**Empirical Novelty And Significance:** 2
**Recommendation:** 5

**Clarity, Quality, Novelty And Reproducibility:**

I believe this paper has large room to improve writing:
1. Informal illustrations. The captions in Figure 1~5 are hard to read. Figures 5, 7, and 8 are not in pdf format. Besides, it is interesting to see four tables in 'Figure 6'.
2. There are many typos in the paper.
3. Section 3.2 is essential, yet it is not clear enough.

**Strength And Weaknesses:**

Strengths:
1. This paper studies an important problem, combinatorial generalization.
2. The authors aim to improve the scalability issue of the previous neural symbolic approaches.

Weaknesses:
1. This paper makes strong assumptions that limit their scope. For example, in the heterogeneous scenario where objects are significantly different in scale (e.g. side length), we cannot directly transfer the knowledge (e.g., offline transition) of a large object to a smaller one.
2. The planning and control algorithms are not clear. For instance, how to search for a reasonable plan without conflicts. I also cannot figure out how to align the entities concretely from only one sentence minimizes the matching cost between z_t and z_g.'. I believe this section is more important than the training details of the submodules. Thus, I recommend that the authors re-organize the paper to emphasize this section more.
3. There are many typos in this paper. I recommend that authors improve their writing, especially in the method section.

**Summary Of The Paper:**

This paper aims to tackle the combinatorial generalization problem in the offline sensorimotor object rearrangement task. By homogenizing the dynamics of different objects, the authors propose a framework named NCS that trains the correspondence matching module and plans a path from the initial state to the goal state in a non-parametric manner. Results on the proposed environments demonstrate that the proposed method significantly outperforms the offline RL baselines.

**Summary Of The Review:**

Overall, the authors touch on an important problem in machine learning. Improving the scalability of the non-parametric methods is also appreciated. However, the assumptions made in the paper are strong and there is much room to improve the writing of this paper.
So I lean towards rejecting this paper.

---

> ### Author Response · Authors · 2022-11-15
> **Response**
>
> We thank reviewer wsHP for their helpful comments, which have helped in our paper in many ways, which we have updated and uploaded. We have addressed their writing concerns with several edits to the writing, as indicated by the blue text in the updated manuscript. We address their additional concerns below.
>
> **strong assumptions**: the reviewer was concerned that we make strong assumptions about the uniformity of scale. However, our method does operate on objects of varying sizes, as can be seen in Figure 5a. The size difference among our largest and smallest objects is at least 2.8x. We have observed that the method sometimes creates multiple nodes in the state transition graph for the same state. This means that there is nothing restricting the method for creating multiple nodes for the same state, but where each node captures an object of a different size.
>
> **Clarity of section 3.2**: we have updated section 3.2 to clarify how the matching cost is implemented. The reason why we used one sentence is that the procedure is indeed quite simple, which is to simply apply Scipy’s Hungarian algorithm implementation (https://docs.scipy.org/doc/scipy-0.18.1/reference/generated/scipy.optimize.linear_sum_assignment.html) to minimize the matching cost (composed of Euclidean distances) among the z’s.
>
> **Figures**: we have updated the manuscript to increase the font size in the captions as well as reformat the tables.

---

### Official Review · Reviewer_vW5u · 2022-11-03

**Confidence:** 4
**Correctness:** 4
**Technical Novelty And Significance:** 3
**Empirical Novelty And Significance:** 3
**Recommendation:** 8

**Clarity, Quality, Novelty And Reproducibility:**

Clarity: well-written paper. Figures can be improved.
Quality: good. analysis from appendix can be moved to the main paper.
Novelty: Visual planning (called object rearrangement in this paper) is actually a very hard task.  I expect the neurosymbolic method introduced in this paper to have a decent impact on the vision+reasoning domain.
Reproducibility: The algorithms are implementation description are useful, but I didn't see any uploaded code.

**Strength And Weaknesses:**

*Strengths*
- Object rearrangement using image inputs is a very interesting problem and I can see a lot of domains where it could be useful: robotics, planning, visual reasoning, navigation / RL to name a few.
- The paper is well written and does a good job of establishing the preliminaries and the method (algo 1, 2, and the figures are especially useful in this regard)
- experiments appear to be extensive on three datasets/environments (although, note that I am not an expert in RL/planning), with recent baselines
- The neuro-symbolic system proposed has the potential to be applicable in various domains, by simply swapping the visual module.

*Weaknesses*
- One weakness (which is fixable) is that IMO it would be better to introduce the problem statement very early on so that readers get a taste of what's to come later.  The introduction is dedicated to mostly explaining the importance and difficulty of object rearrangement, without specifying what input-output spaces are being considered -- this comes very late.
- It would be interesting to see the method being applied to real world datasets -- i.e. images with real objects (see my suggestion 2 for prospective datasets). Perhaps in such settings, more sophisticated feature learning / entity abstraction methods would need to be used.
- ablation study/analyses to study each module could enhance the evidence for the efficacy of the method

*Suggestions*
1. Increase font size in all figures. Tables are fine.
2. References: there are two related papers that fall under the problem formulation given in Sec 2 Page 2:
  - https://openaccess.thecvf.com/content_CVPRW_2019/papers/Vision_Meets_Cognition_Camera_Ready/Gokhale_Cooking_With_Blocks__A_Recipe_for_Visual_Reasoning_on_CVPRW_2019_paper.pdf This paper is about an object rearrangement task in the "blocksworld" setting but with real images. This one seems to be very similar to _"block-stack"_ from this paper.
  - https://www.ecva.net/papers/eccv_2020/papers_ECCV/papers/123560324.pdf procedural planning in videos -- this paper involves complex actions beyond simple rearrangement, but in real-world videos like cooking etc.


**Summary Of The Paper:**

This paper is about the task of object rearrangement. The paper proposes a neurosymbolic approach for solving this problem.
The method is described as a hierarchical abstraction approach-- object representations are learned, and a factorized transition graph is constructed over these representations.  The method, "Neural Constraint Satisfaction" disentangles representation learning from planning and control, therefore leading to a modularized approach.  Experiments are performed on three block-style datasets.


**Summary Of The Review:**

An effective neurosymbolic method for object re-arrangement, effective across many domains.  Real-world applicability (i.e. with complex images) has not been explored -- all the experimental settings are in synthetic environments, which in my opinion, limits the impact of this paper

---

> ### Author Response · Authors · 2022-11-15
> **Response**
>
> We thank reviewer vW5u for their generally positive assessment of our work and its writing. The reviewer's comments have substantially improved the manuscript, which we have updated and uploaded. We address their additional comments in detail below:
>
> **introducing problem statement in the intro**: The reviewer asked for the problem statement to be introduced in the intro. Our original submission had introduced the problem statement in the last sentence of the second paragraph of the introduction. We have revised that sentence to make the problem statement clearer: *“We can formulate this as an offline goal-conditioned reinforcement learning (RL) problem, where the agent is trained to model a dataset of sensorimotor interactions and is evaluated on outputting a sequence of actions for rearranging objects specified in the input image to satisfy constraints depicted in a goal image.”*
>
> **Real world datasets**: We agree with the reviewer that extending the method to real world datasets would be valuable future work. Our focus is specifically to contribute a framework for applying object-centric methods to interactive rearrangement tasks, so we focus on simulated environments where we can properly evaluate task completion given the simulator state, thereby enabling us to focus the paper specifically on the rearrangement problem itself without needing to consider the open research problem of real world evaluation, which is orthogonal to our work’s focus. We also note that prior object-centric works still primarily operate in simulation for passive image and video data, but our work is one of the few that extend existing methods to the interactive setting. Furthermore, our main contribution is our factorized planning algorithm which is orthogonal to what object-centric representation learning method is used. As object-centric representation learning methods mature (e.g. future iterations of SLATE), we would expect our method to be more readily applicable to real world data.
>
> **Ablations and Analysis**: In our submission we have analyzed and ablated our method in nine ways, and we have clarified this in the introductory paragraph of section 4.  The first two ways are quantitative comparisons between NCS (which performs factorized graph search over factorized latents) and two baselines: MGS performs monolithic graph search by treating each set of latents monolithically and MPC performs receding horizon planning over the factorized latents. The third way is the qualitative visualization of the state clusters NCS extracts (Figure 7) and the fourth way is the qualitative visualization of the algorithm NCS executes during test time (Figure 4). The fifth and sixth ways are presented in Figure 8, where we analyze how NCS performs with varying levels of action noise as well as varying horizon lengths. The last three ways are presented in Figure 11, we analyzed the effect of changing the number of batches, number of slots, and number of clusters for constructing the transition graph.

---

> > ### Comment · Reviewer_vW5u · 2022-11-19
> > **Thanks**
> >
> > Thanks for the response.
> > - I think the changes made to the introduction, especially the changes on page 2, will make the paper more transparent and easier to grasp for the reader.
> > - I understand that extension to real-world datasets is not the current focus of this paper.  Nevertheless, I believe that there should be an open-ended discussion in the paper (perhaps in the conclusion?) about what the approaches / associated challenges would be to apply this method in the future to more complex domains.  With that in mind, including references to papers (for instance the ones mentioned in my review) that have also addressed the rearrangement problem (or variants) (with limited success but in a real-world/complex setting) is important.
> > - About ablations:  yes I understand that the ablations are in the appendix.  That is why in my review, under "Clarity, Quality, Novelty And Reproducibility:" I have suggested moving important analyses to the main paper.

---

> > > ### Author Response · Authors · 2022-11-19
> > > **Thank you for your response**
> > >
> > > We have updated the discussion section in a new revision of the paper with a discussion on the additional challenges that need to be overcome to scale our method to real world environments and have cited the papers you had referenced.

---

### Decision · Program_Chairs · 2023-01-20

**Decision:**

Accept: poster

**Justification For Why Not Higher Score:**

The limited current scope and the NCS’s design appears to rely too heavily on certain strong assumptions that cater to the synthetic rearrangement environments it is demonstrated on.

**Justification For Why Not Lower Score:**

The problem formulation is novel, and after revision, the draft is well presented in its current form.



**Metareview: Summary, Strengths And Weaknesses:**

This paper is about the task of object rearrangement. The paper proposes a neurosymbolic approach for solving this problem. The method is described as a hierarchical abstraction approach-- object representations are learned, and a factorized transition graph is constructed over these representations. Results on the proposed environments demonstrate that the proposed method significantly outperforms the offline RL baselines.

+ The paper presents an interesting new problem;
+ The method proposed has wide potential;

The reviewers suggested to clarify the problem statement, and it has been taken care of. I agree with Reviewer vW5u's suggestions to make the work more transparent, easier for ICLR's audiences to follow.

**Note From Pc:**

if the above contains the word "oral" or "spotlight" please see: "oral" presentation means -> notable-top-5% and "spotlight" means -> notable-top-25%. As stated in our emails, we are disassociating presentation type from AC recommendations

**Summary Of Ac-Reviewer Meeting:**

N/A